# Impact of Playing Position on Competition External Load in Professional Padel Players Using Inertial Devices

**DOI:** 10.3390/s25030800

**Published:** 2025-01-29

**Authors:** Ricardo Miralles, José F. Guzmán, Jesús Ramón-Llin, Rafael Martínez-Gallego

**Affiliations:** 1Research Group of the Padel Federation of the Valencian Community, University of Valencia, El Llano del Real, 46010 Valencia, Spain; rimiser@alumni.uv.es; 2Research Group on Sports Technique and Tactics, University of Valencia, 46010 Valencia, Spain; jesus.ramon@uv.es (J.R.-L.); rafael.martinez-gallego@uv.es (R.M.-G.)

**Keywords:** accelerometer, decelerometer, EPTS, sports performance, racket sport, elite, padel

## Abstract

Padel is a racket sport that has grown internationally, both in the number of players and in the number of competitions. Inertial measurement devices enable a comprehensive analysis of competitive load in padel by providing kinematic variables that enhance players’ performance in this discipline. This study aimed to analyse the external load variables recorded with an inertial device in elite padel players, comparing metrics based on the players’ positions (left and right sides of the court). A total of 83 players were monitored during 23 matches of the professional circuit. The results revealed specific load metrics, including distance covered, frequency of accelerations and decelerations per hour, maximum speeds reached, and acceleration profiles relative to distance covered, which were all measured using the Wimu Pro™ device. Left-side players showed more frequent accelerations and decelerations per hour compared to right-side players. The results of this study will, on one hand, enable the adjustment of new specific parameters for professional padel training, such as acceleration and deceleration profiles, player load, and distances covered at explosive speeds. On the other hand, the results will provide a more objective evaluation of padel players’ performance based on their positions.

## 1. Introduction

Padel has grown rapidly as a global sport [1], with the Premier Padel circuit hosting 25 tournaments across 18 countries on 5 continents [2]. The increase in the number of players and the global reach of the professional padel circuit have led to a rise in research and publications focused on padel [3]. Consequently, several studies emphasise the importance of conducting competition-based research, as the findings provide valuable insights for developing more specific and effective training methodologies [4].

Research on padel mainly focuses on performance indicators and competition load [5,6,7]. Player load refers to the cumulative exertion an athlete endures during a given activity. Measures of player load are generally categorised as either internal or external [8,9,10]. Internal training load is defined as the biological stimuli imposed on the athlete during competition [11].

In analysing internal load in padel, most studies have utilised heart rate (HR) as a key parameter [12,13,14,15,16,17], with findings indicating that padel players typically operate at approximately 84.90% of their maximum HR [18].

External load analysis in padel has primarily addressed temporal structure [12,19,20,21], player movement patterns [22,23], and technical–tactical parameters [24,25,26]. In recent years, Electronic Performance Tracking Systems (EPTSs) have become essential tools for workload monitoring in team sports, particularly football [27]. In this context, inertial devices—often referred to as Inertial Measurement Units and incorporating various sensors such as accelerometers, gyroscopes, magnetometers, and GNSSs—are increasingly utilised [28,29]. EPTSs enable the real-time analysis of physical and tactical parameters, providing teams with critical data and immediate feedback from the field. These systems allow for the analysis of kinematic variables, such as acceleration, deceleration, distance covered, explosive distance, number of steps, maximum speed, average speed, speed zones, and neuromuscular variables, including impacts, Player Load, and Metabolic Power, among others.

Furthermore, playing position has been shown to significantly impact the performance profile of players in terms of distance travelled, number and type of strokes, shot efficiency, and psychological characteristics [23,30,31]. In terms of distance travelled, Ramón-Llin [23] found that players on the left side, also known as left-side or forehand players, covered significantly more distance than their partners when both were right-handed. However, these differences diminished when the right-side player was left-handed.

From a technical–tactical perspective, players with greater power and more effective power shots [25,32] are often positioned on the left side if they are right-handed and on the right side if they are left-handed. Nevertheless, although left-handed players appear more effective in executing power shots, they tend to commit more errors when hitting the ball after a wall rebound [25,32]. Right-side players, in contrast, typically perform more lobs and commit fewer errors [31,32]. Ramón-Llin et al. [32] observed that right-side players are more likely to hit parallel strokes, whereas left-side players favour cross-court strokes and achieve more winning shots. This suggests a specialisation where left-side players act as “scorers” while right-side players serve as “defenders” [5].

Regarding psychological factors, left-side players exhibit higher levels of somatic anxiety and self-confidence before competition compared to training matches [33]. Concerning physical variables, Ortega-Zayas et al. [34] reported slightly higher values across all analysed physical variables for players positioned on the left side.

However, despite the professionalisation and international growth of the padel circuit, a literature review reveals a lack of data on acceleration profiles, explosive distance, and player load in the padel based on court position. Therefore, the objective of this study was to analyse external load variables recorded with an inertial device in elite padel players, with a specific focus on court positioning.

## 2. Materials and Methods

### 2.1. Sample

A total of 84 professional male players, aged between 20 and 44 years, were included in this study. This corresponds to a sample size of 84 subjects, calculated to compare two related samples in a one-tailed test with an alpha level of 0.05, a statistical power of 0.90 (1-β), and an effect size of 0.33 (G*Power). All participants held a professional ranking on the World Padel Tour, ranging from 40 to 216 globally. The matches analysed comprised 9 finals and 14 semi-finals from the Gold 24 World Padel Tour Next Season 2021–2022. All players participated voluntarily, providing informed consent before the commencement of the study. A Regional Padel Federation was responsible for obtaining the necessary permits to record the facilities and the players. This study adhered to the ethical principles outlined in the Declaration of Helsinki.

### 2.2. Instrument

EPTSs were employed in this study to analyse and record data on player performance. Specifically, the Wimu Pro™ device (RealTrack Systems, Almería, Spain) was utilised. This device is a hybrid EPTS, integrating both Local Positioning System (LPS) and Global Navigation Satellite System (GNSS) technology within a single unit. It enables performance monitoring through GNSSs for outdoor facilities and supports indoor facilities with limited GNSS capabilities. Over the past decade, this technology has undergone continuous development, enhancing parameters such as sampling frequency, device positioning, satellite signal strength, and data integrity [35]. In this study, all data recordings were conducted in outdoor courts. The Wimu Pro™ has demonstrated validity in positioning and tracking applications [36,37]. A GNSS receiver with a sampling frequency of 18 Hz was used for data collection.

### 2.3. Procedure

This study analysed 9 finals and 14 semi-finals of the Gold 24 World Padel Tour Next Season 2021–2022. Data were collected from each player in every match, with each participant providing informed, signed consent. Before the general warm-up, players were fitted with a tracking vest without the Wimu Pro™ device. After completing specific warm-up exercises, the Wimu Pro™ device was inserted into the vest and activated, and data recording began with the player’s first serve. Data recording ceased upon completion of the last point of the match.

During each match, the WimuPro™ device recorded over 250 variables. The Svivo Server software (Version 2022.923.0.0) enabled real-time data visualisation. After each match, the data were further analysed using the Spro software and stored in the Wimu Cloud alongside all match results.

### 2.4. Variables Analysed

The variables analysed in this study are external load variables, which are detailed below. The classification of the dependent variables is based on Miralles et al. [38], with the aim of facilitating their organisation into ‘simpler’ and more traditional variables, as well as more complex and novel ones:Variables related to load volume:-Duration: Refers to the total length of the match, measured in seconds.-Distance: Represents the total distance travelled during a match, measured in metres.-Relative distance: Refers to the total distance covered during a match, normalised per hour of play (m/h).Variables related to classic load volume and intensity (traditional metrics):-Maximum speed: The highest speed reached by the player during the match, measured in km/h.-Acceleration + 1: The total number of accelerations exceeding 1.12 m/s^2^.-Relative acceleration: The total number of accelerations equal to or greater than 1.12 m/s^2^, normalised per hour of match (m/h).-Max acceleration: The maximum acceleration achieved, measured in m/s^2^.-Deceleration + 1: The total number of decelerations at −1.12 m/s^2^.-Relative deceleration: The total number of decelerations at −1.12 m/s^2^, normalised per hour.-Max deceleration: The maximum deceleration achieved, measured in m/s^2^.Variables related to novel loading volume and intensity (advanced kinematic variables):-Player load: The vector sum of accelerations across the three orthogonal axes (vertical, anteroposterior, and lateral). This variable is used to assess neuromuscular load across different athletes [39] and is calculated as the vector sum of accelerations along three axes (anteroposterior, lateral, and vertical), including both accelerations and decelerations. A cumulative measure and an intensity measure are analysed, offering insights into the rate of stress exerted on the body over a specified period:
$$Player\;Load=\frac{\sqrt{{\left(a_{y1}-a_{y-1}\right)}^{2}+{\left(a_{x1}-a_{x-1}\right)}^{2}+{\left(a_{z1}-a_{z-1}\right)}^{2}}}{100}$$
-Explosive distance: The total distance covered with an acceleration greater than 1.12 m/s^2^.-Relative Explosive distance: The total distance covered with an acceleration greater than 1.12 m/s^2^ during a match, normalised per hour of session duration (m/h).-HSR Relative: The distance travelled (metres) at speeds above the player’s individual threshold (75.5% of their maximum speed), calculated based on their historical maximum speed. For instance, if a player’s recorded maximum speed is 15.21 km/h, their HSR absolute threshold would be reached whenever they exceed 11.4 km/h.Additionally, an acceleration profile is constructed to include the following:-Accelerations 1–2 m: The number of accelerations performed covering a distance between 1 and 2 m.-Accelerations 2–3 m: The number of accelerations performed covering a distance between 2 and 3 m.-Accelerations > 3 m: The number of accelerations performed covering a distance greater than 3 m.-Decelerations 1–2 m: The number of decelerations performed covering a distance between 1 and 2 m.-Decelerations 2–3 m: The number of decelerations performed covering a distance between 2 and 3 m.-Decelerations > 3 m: The number of decelerations performed covering a distance greater than 3 m.
Independent variable:-Player position: The side from which the player starts the point (right or left side).

### 2.5. Data Analysis

For data analysis, the statistical software R Studio (R-Tools Technology) was used for both analyses and graph creation. The median and interquartile range were calculated for descriptive purposes. Initially, Kolmogorov–Smirnov (K-S) tests were conducted to assess normality. Wilcoxon tests were performed to compare load variables based on playing position. Effect size was calculated using Pearson’s r, classified as follows: 0.5 as a large effect, 0.3 as a medium effect, and 0.1 as a small effect. Significance was set at *p*-values ≤ 0.05.

## 3. Results

### Analysis by Court Position

Table 1 presents the movement of external load variables according to court position. Left-side players showed higher median values across most variables, close to significant differences in relative accelerations (Mdn_right_ = 377.54 vs. Mdn_left_ = 413.54; *p* = 0.05; r = 0.22). Other differences, such as relative decelerations (Mdn_right_ = 333.59 vs. Mdn_left_ = 373.96; *p* = 0.06; r = 0.20), approached significance. Although this difference did not reach statistical significance, it suggests a potential higher deceleration demand on the left side.

Figure 1 illustrates the number of accelerations and decelerations per hour across various distance ranges based on court position. For accelerations, as indicated before, players on the left side performed a significantly higher number of accelerations per hour (Mdn = 413.54) compared to those on the right side (Mdn = 377.54). Although the differences were not statistically significant for specific distance ranges, left-side players generally showed higher counts across all acceleration distances. Additionally, approximately 80% of all accelerations occurred within the 1–2 m range (Mdn_left_ = 290.24; Mdn_right_ = 263.90). Regarding decelerations, although no statistically significant difference was found, a trend was observed with left-side players performing more decelerations per hour (Mdn = 373.96) than right-side players (Mdn = 333.59; *p* = 0.06; r = 0.24). As with accelerations, approximately 80% of decelerations occurred within the 1–2 m range (Mdn_left_ = 290.24; Mdn_right_ = 263.90).

## 4. Discussion

Technological advancements in recent years have significantly enhanced the optimisation of athlete performance [40], with an increasing reliance on GPS devices to describe the physical profiles of athletes across sports [41]. In the context of padel, adopting EPTSs is essential to understanding the physical demands of players during competition, particularly as padel is characterised as a sport with an intermittent nature [42]. The present study aimed to describe competitive load characteristics based on playing position, using an inertial EPTS device to analyse variables related to external load volume and intensity. Furthermore, an acceleration profile was established for matches involving professional players.

The analysis of court position shows that left-side players exhibited a higher rate of acceleration relative to match duration compared to right-side players, a trend toward higher median values across nearly all load-related variables, except for high-speed running relative, compared to right-side players. Notably, left-side players showed a significantly higher relative number of accelerations per hour than right-side players, reflecting a small to moderate effect. Although other variables did not reach statistical significance, a trend was observed for relative decelerations per hour, with left-side players displaying higher values than their right-side counterparts.

Left-side players face greater physical demands due to technical–tactical actions requiring more mobility and court coverage. They engage more frequently in play, executing more volleys, smashes, and wall rebounds compared to right-side players [43]. This heightened involvement necessitates rapid movements, leading to a greater number of accelerations and decelerations, higher speeds, and increased distances covered, which is ultimately reflected in higher values for player load and explosive distance. These results align with previous findings by Ramon-Llin [23], who reported that players on the left side generally covered greater distances across all levels of play. In this sense, Ramón-Llin [23] compared the distance travelled per point (10.74 m for the player on the right side versus 11.43 m for the player on the left side), while in our study, we have compared the total distance per match (3381 m for the player on the right side versus 3476 m for the player on the left side). Differences were more pronounced when both players in the pair were right-handed, while distance travelled was comparable when the left-side player was right-handed, and the right-side player was left-handed.

An important characteristic of padel is the rapid shift to offensive positions near the net immediately following the serve [44]. However, recent studies suggest that the advantage of reaching these offensive positions decreases after 6–8 strokes [45,46]. This implies that professional padel favours players who can effectively secure points with a limited number of attacking actions per rally. Consequently, left-side players not only engage more frequently in play with a diverse range of technical–tactical actions but also demonstrate higher values for accelerations, decelerations, distances covered, maximum speeds, and player load. These results provide a comprehensive conditional profile that supports the notion that left-side players adopt a more active and physically demanding role during matches [42].

The findings of this study offer several practical implications for coaches, trainers, and players seeking to optimise performance in professional padel. Firstly, the data suggest that left-side players have distinct physical and technical demands, which require targeted conditioning to support higher rates of accelerations and decelerations and the capacity to cover more ground efficiently. Conditioning programmes for left-side players could focus on agility, explosiveness, and anaerobic endurance to sustain the intense, repetitive bursts of movement required during matches. As well as adding lower-body eccentric, bilateral or unilateral exercises, considering the different movements on the track. Additionally, coaches could use these insights to develop tactical strategies that play to each player’s strengths. Therefore, players on the left side need to be managed differently in terms of recovery and rest variables to prevent injuries. This could include a personalised approach to stretching sessions, specifically assessing muscle fatigue in left-sided players and planning more frequent rest or active rest periods to promote faster recovery. These strategies can help reduce the risk of overload and improve long-term performance [47]. These factors will be crucial for coaches to address, tailoring them according to each player’s specific load. For example, if left-side players naturally engage in more attacking actions and cover more court areas, strategies could be designed to maximise their involvement in offensive plays, with right-side players positioned to support defensively or manage prolonged rallies. Training could incorporate drills that mimic match scenarios, with left-side players practising rapid recovery between intense bursts of movement and developing the endurance to maintain high performance under these conditions. Furthermore, these results highlight the importance of load monitoring tailored to each player’s court position. By tracking load-specific metrics such as player load, accelerations, and decelerations, coaches and sports scientists can better manage training intensities and avoid overtraining or injury. This is particularly relevant given that left-side players might be at greater risk of physical strain due to their higher engagement in physically demanding actions. Adjustments in workload could be implemented based on the data, allowing players on the left to receive rest periods or modified training sessions as needed.

While these findings provide valuable insights into competitive load profiles, further research is needed to fully explore positional demands in padel. The main limitation of the study is that the sample size is sufficient to detect large (d = 0.8) and medium (d = 0.5) effect sizes but not small ones (d = 0.2) cause the study was able to detect differences starting from an effect size of d = 0.33. Future studies could investigate the role of left-handed players [28]), as their presence may influence dynamics and strategies on both sides of the court. Additionally, the impact of serving strategies—whether conventional or Australian—on load and court positioning would be an interesting area to explore. It would be very interesting to study how external load evolves when comparing the first and second halves of the match. In our study, we did not perform measurements on professional female padel players, as this will be the subject of future studies, as well as performing internal load measurements, a clear example could be HR. It will also be the purpose of future research to analyse the kinematic and dynamic parameters in movement techniques. Finally, analysing the anthropometric characteristics of players in relation to their playing position could provide insights into how physical traits affect movement patterns, tactical roles, and overall performance on the court.

## 5. Conclusions

The study was one of the first to compare novel external loading variables such as player load and acceleration profiles considering position. The findings of this study indicate that left-side professional padel players exhibit higher values across most external load variables compared to right-side players. Notably, they showed close to significantly higher relative acceleration rates per hour, reflecting a greater physical load.

These results underscore the importance of position-specific conditioning in professional padel. Coaches and trainers should consider the distinct physical demands of left-side players when designing targeted training programmes, ensuring their preparation is tailored to the more active and physically demanding role they assume during matches.

## Figures and Tables

**Figure 1 sensors-25-00800-f001:**
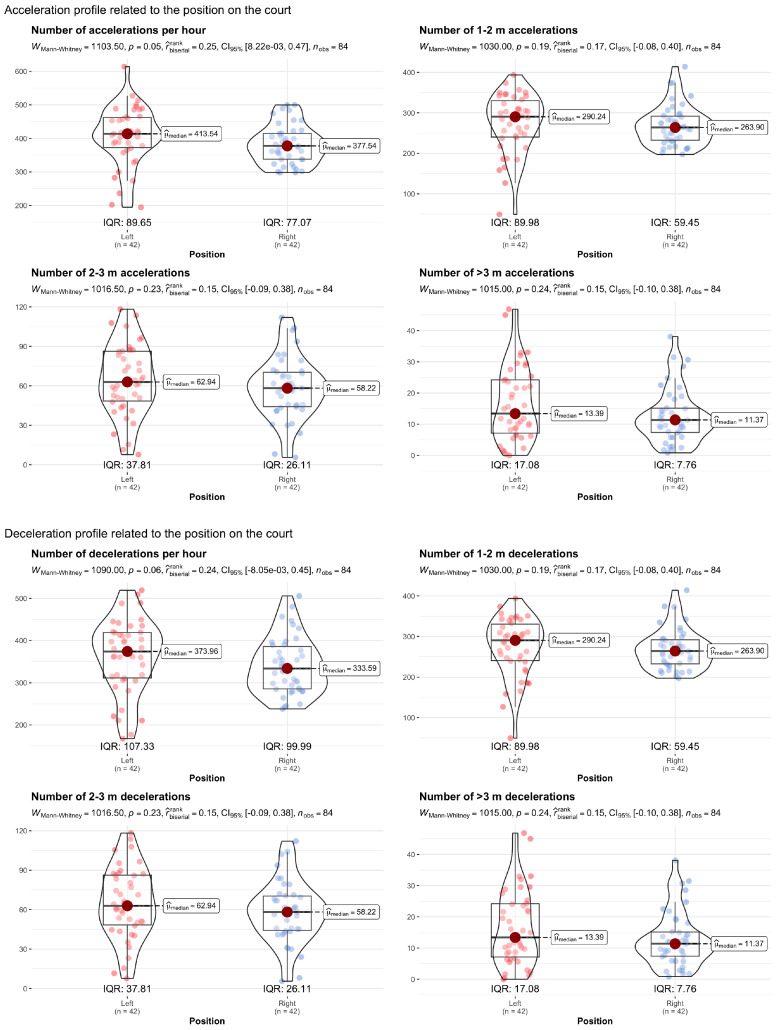
Acceleration and deceleration profiles by court position in elite padel players.

**Table 1 sensors-25-00800-t001:** Court position and movement variables in elite padel players.

Variable	Right Side Median (IQR)	Left Side Median (IQR)	Z	*p*	r
Volume variables					
Duration_(s)	5226.00 (3341.25)	5057.00 (3341.25)	840.50	0.71	0.04
Distance (m)	3381.51 (1948.99)	3474.61 (2185.10)	892.0	0.93	0.01
Distance_Rel (m)	2370.07 (551.16)	2482.47 (453.56)	1047.0	0.14	0.16
Volume and intensity classical variables					
Max_Speed_(km/h)	15.18 (3.27)	15.40 (3.13)	991.0	0.33	0.11
Accelerations + 1	614.00 (384.00)	579.00 (328.50)	894.0	0.92	0.01
Accelerations_Rel	377.54 (77.07)	413.54 (89.65)	1103.5	0.05	0.22
Max_Acceleration (m/s^2^)	4.79 (1.21)	4.54 (0.73)	799.0	0.46	0.08
Decelerations + 1	516.50 (316.25)	534.00 (352.75)	911.5	0.80	0.03
Decelerations_Rel	333.59 (99.99)	373.96 (107.33)	1090.0	0.06	0.20
Max_Deceleration (m/s^2^)	−4.46 (0.97)	−4.56 (1.12)	811.5	0.53	0.07
Volume and intensity novel variables					
Player_Load_(a.u.)	53.88 (33.35)	56.41 (36.54)	814.0	0.55	0.07
Explosive_Distance_(m)	387.24 (207.94)	404.59 (257.96)	915.0	0.77	0.03
Explosive_Distance_Rel	42.40 (91.37)	272.63 (125.06)	1029.0	0.19	0.14
HSR_Rel_	8.37 (17.30)	7.07 (11.91)	856.0	0.82	0.03

## Data Availability

Data are unavailable due to privacy.

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
