# Peer review of "Impact of Playing Position on Competition External Load in Professional Padel Players Using Inertial Devices"

_sensors, 2025, doi:10.3390/s25030800_

Round 1
Reviewer 1 Report
Comments and Suggestions for Authors
Dear Authors,
Your manuscript titled "Impact of Playing Position on Competition Load in Professional Padel Players Using Inertial Devices." is well-structured and provides valuable insights into external load differences based on player positions in padel. Below, I provide detailed feedback to enhance the clarity, coherence, and scientific rigor of your manuscript. My suggestions address the abstract, introduction, methodology, results, discussion, and conclusions, with a focus on improving readability and ensuring the data's implications are well-justified.
Abstract:
· The opening sentence is too broad ("enhance athlete performance across various sports"). This may cause confusion since the study is specifically focused on padel. It is suggested to include a mention of the growing popularity of this sport.
· Information regarding the type of device used and the range of data measured is not provided, which could give better context.
· The final sentence ("offering data-driven guidance to optimize competitive performance") is somewhat generic. It would be more useful to link the results specifically to practical implications for coaches or players.
Introduction:
· The term "Electronic Performance Tracking Systems (EPTS)" appears three times in the manuscript; please revise to avoid redundancy.
· Moreover, "Inertial Measurement Units (IMUs)" is mentioned only once in the text, so abbreviating it seems unnecessary.
Methodology:
· It is always valuable to include information on which ethics committee approved the study (in this case, it should be CEICA) and the corresponding code.
· In the section on analyzed variables, the differentiation between internal and external load might be better placed in the introduction.
· The methodology should focus on external load, as that is precisely what is being measured.
· Was a sample size calculation performed? Is the sample representative? With a larger sample, perhaps more significant differences could have been observed.
· The division between "classic volume" and "novel volume and intensity" might be confusing for less familiar readers. Simplifying these terms or providing a summary table explaining the differences between these categories is suggested.
· Thresholds (>1.12 m/s²) for analyzing accelerations and decelerations are established, but it is not explained how these values were defined. It is suggested to justify the selection of these thresholds based on previous studies or existing literature.
· Figure 1 is very well-designed.
· It would be very interesting to study how external load evolves when comparing the first and second halves of the match. If it is not feasible to do so, it would be appropriate to mention this in future research and/or limitations.
Results:
· In the opening sentence: "Table 1 presents the movement variables according to court position. Players positioned on the left side demonstrated higher median values than those on the right side across most variables." It is suggested to specify that these were external load variables.
Discussion:
· A paragraph on limitations is necessary, including issues like the exclusion of female athletes, the lack of internal load measurements (e.g., heart rate), and environmental factors such as the effect of score, court, temperature, opponents, and players' anthropometric characteristics. Furthermore, it is suggested to allow access to the database by anonymizing or pseudonymizing it.
· While practical implications are offered, it is not discussed how these differences in load could influence injury incidence or long-term performance. It is suggested to expand the discussion on the potential impact of higher physical demands on recovery and injury prevention, especially for left-side players.
· Some important findings did not reach statistical significance, but their discussion could benefit from a more critical analysis of their practical relevance. It is suggested to explain more clearly why these results, though not statistically significant, remain important from a technical-tactical perspective, to "compensate" for the lack of significance.
· Here: "The present study aimed to describe competitive load characteristics based on playing position, using an inertial EPTS device to analyze variables related to load volume and intensity." It is suggested to specify that the focus is on external load.
· It is suggested to clarify that the results show "left-side players exhibited a higher rate of accelerations relative to match duration compared to right-side players," along with "a trend toward higher median values across nearly all load-related variables, except for high-speed running relative, compared to right-side players," instead of "The analysis of court position revealed that players on the left side exhibited higher median values across nearly all load-related variables, except for high-speed running relative, compared to right-side players." Since there were no statistically significant differences, it is necessary to be careful with the terms used. Similarly, here: "The analysis of court position revealed that players on the left side exhibited higher median values across nearly all load-related variables, except for high-speed running relative, compared to right-side players."
· Here: "These results align with previous findings by Ramon-Llin (20), who reported that players on the left side generally covered greater distances across all levels of play," it would be good to compare the numerical data from that study with the present one.
· It is suggested to add lower-body eccentric exercises and discuss the application of bilateral and unilateral exercises, considering padel movements: "Conditioning programs for left-side players could focus on agility, explosiveness, and anaerobic endurance to sustain the intense, repetitive bursts of movement required during matches."
Conclusions:
· Again, in the conclusions, it is necessary to mention that the study focuses on external load.
· The conclusions are too lengthy, and some parts belong to the discussion, such as the information starting with "These results align with the understanding." My proposed conclusion to give the authors an idea is as follows:
· "The findings of this study indicate that left-side professional padel players exhibit higher values across most external load variables compared to right-side players. Notably, they showed significantly higher relative acceleration rates per hour, reflecting a greater physical load. Additionally, a trend toward higher relative deceleration values was observed for left-side players.
· These results underscore the importance of position-specific conditioning in professional padel. Coaches and trainers should consider the distinct physical demands of left-side players when designing targeted training programs, ensuring their preparation is tailored to the more active and physically demanding role they assume during matches."

The English language in the manuscript is generally clear but could benefit from minor revisions to improve readability and precision.
Author Response
REVIEWER 1
We would like to thank the reviewer for their positive comments. All proposed changes have been highlighted in red in the manuscript, and specific responses to each of the reviewer’s suggestions are provided below.
Abstract:
- The opening sentence is too broad ("enhance athlete performance across various sports"). This may cause confusion since the study is specifically focused on padel. It is suggested to include a mention of the growing popularity of this sport.
Thank you for your comment. It has been rewritten and added “Padel is a racket sport that has grown internationally, both in the number of players and in the number of competitions”. Intead of “athlete performance”we rewrite “padel players performance”
- Information regarding the type of device used and the range of data measured is not provided, which could give better context.
We have included information on the device used reporting “using the Wimu Pro™ device”
- The final sentence ("offering data-driven guidance to optimize competitive performance") is somewhat generic. It would be more useful to link the results specifically to practical implications for coaches or players.
Thanks for you comment. We have added at the end of the abstract: “The results of this study will, on the one hand, enable the adjustment of new specific parameters for professional padel training, such as acceleration and deceleration profiles, player load, and distances covered at explosive speeds. On the other hand, the results will provide a more objective evaluation of padel players' performance based on their positions
The wording of the abstarct in English has been revised to make it more precise.
Introduction:
- The term "Electronic Performance Tracking Systems (EPTS)" appears three times in the manuscript; please revise to avoid redundancy.
Thank you for your comment. The first time ‘Electronic Performance Tracking Systems (EPTS)’ has been retained for subsequent mentions only to indicate ‘EPTS’. We have made the correction of this point in line; 48, 88, 90,208 211, 274 of the manuscript.
- Moreover, "Inertial Measurement Units (IMUs)" is mentioned only once in the text, so abbreviating it seems unnecessary.
This correction was made in line with the manuscript 46, we thank you.
Methodology:
- It is always valuable to include information on which ethics committee approved the study (in this case, it should be CEICA) and the corresponding code.
As indicated in the manuscript and also indicated to the editors of the magazine, the data collection process was carried out by the Paddle Federation. It was a Regional Padel Federation, which was responsible for obtaining the permits to carry out the records and recordings, both of the facilities and of the players. Therefore, the University’s ethics committee is not required to approve this research. We have added that” A Regional Padel Federation was responsible for obtaining the necessary permits for recording both the facilities and the players.”
- In the section on analyzed variables, the differentiation between internal and external load might be better placed in the introduction.
We appreciate this correction; we have made the correction of this point in line 35. “Measures of player load are generally categorised as either internal or external (8,9,10). Internal training load is defined as the biological stimuli imposed on the athlete during competition (11)”
- The methodology should focus on external load, as that is precisely what is being measured.
Thank you. We agree that all the variables analysed and described in the methodology are externally loaded.
- Was a sample size calculation performed? Is the sample representative? With a larger sample, perhaps more significant differences could have been observed·
Thank you for your comment. In the methodological part it has been added in the sample that ‘A total of 84 male professional players, aged between 20 and 44 years, were included in this study. This corresponds to a sample size of 81 subjects, calculated to compare two related samples in a one-tailed test with an alpha level of 0.05, a statistical power of 0.90 (1 - β), and an effect size of 0.33 (G*Power).
The main limitation of the study is that the sample size is sufficient to detect large (d=0.8) and medium (d=0.5) effect sizes but not small ones (d=0.2). The study was able to detect differences starting from an effect size of d=0.33.
The division between "classic volume" and "novel volume and intensity" might be confusing for less familiar readers. Simplifying these terms or providing a summary table explaining the differences between these categories is suggested.
We appreciate this correction; some comments have been made in the manuscript to clarify the terms set forth above. The classification of the variables is based on Miralles et al (2025) with the intention of facilitating the structuring into simpler and more traditional variables to more complex and novel variables.
- Thresholds (>1.12 m/s²) for analyzing accelerations and decelerations are established, but it is not explained how these values were defined. It is suggested to justify the selection of these thresholds based on previous studies or existing literature.
This value has been used with specific studies with paddle tennis. Specifically, the article is the following;
Miralles, R., Martínez-Gallego, R., Guzmán, J., & Ramón-Llin, J. (2025). Movement patterns and player load: insights from professional padel. Biology of Sport, 42(1), 163-169.
Figure 1 is very well-designed.
- It would be very interesting to study how external load evolves when comparing the first and second halves of the match. If it is not feasible to do so, it would be appropriate to mention this in future research and/or limitations.
We have added this appreciation in future research. We appreciate your comment.
Results:
- In the opening sentence: "Table 1 presents the movement variables according to court position. Players positioned on the left side demonstrated higher median values than those on the right side across most variables." It is suggested to specify that these were external load variables.
We have added this appreciation in the manuscript in methodology “The variables analysed in this study are external load variables, which are detailed below” and in the results “Table 1 presents the movement external load variables…”
Discussion:
- A paragraph on limitations is necessary, including issues like the exclusion of female athletes, the lack of internal load measurements (e.g., heart rate), and environmental factors such as the effect of score, court, temperature, opponents, and players' anthropometric characteristics.
We have added this purpose in the manuscript; “In our study, we did not perform measurements on professional female padel players, as this will be the subject of future studies, as well as performing internal load measurements, a clear example could be HR.”.
- Furthermore, it is suggested to allow access to the database by anonymizing or pseudonymizing it
We are going to attach the anonymised database
- While practical implications are offered, it is not discussed how these differences in load could influence injury incidence or long-term performance. It is suggested to expand the discussion on the potential impact of higher physical demands on recovery and injury prevention, especially for left-side players.
We appreciate this correction; we have added in the manuscript this text “Therefore, players on the left side should be managed differently in terms of recovery and rest variables to prevent injuries. These factors will be crucial for coaches to address, tailoring them according to each player's specific load.”.
- Some important findings did not reach statistical significance, but their discussion could benefit from a more critical analysis of their practical relevance. It is suggested to explain more clearly why these results, though not statistically significant, remain important from a technical-tactical perspective, to "compensate" for the lack of significance.
Please, if we have left out any ideas in the modifications made in the results and discussion, we will be happy to read the reviewer's proposal.
- Here: "The present study aimed to describe competitive load characteristics based on playing position, using an inertial EPTS device to analyze variables related to load volume and intensity." It is suggested to specify that the focus is on external load.
Thank for you comment. We have added this appreciation in the manuscript. “The present study aimed to describe competitive load characteristics based on playing position, using an inertial EPTS device to analyse variables related to external load…”
- It is suggested to clarify that the results show "left-side players exhibited a higher rate of accelerations relative to match duration compared to right-side players," along with "a trend toward higher median values across nearly all load-related variables, except for high-speed running relative, compared to right-side players," instead of "The analysis of court position revealed that players on the left side exhibited higher median values across nearly all load-related variables, except for high-speed running relative, compared to right-side players." Since there were no statistically significant differences, it is necessary to be careful with the terms used. Similarly, here: "The analysis of court position revealed that players on the left side exhibited higher median values across nearly all load-related variables, except for high-speed running relative, compared to right-side players."
Thank you for your input. You are right. Instead of indicating that there were significant differences we have indicated that ‘it was close to significant differences’ for the p value=0.05.
- Here: "These results align with previous findings by Ramon-Llin (20), who reported that players on the left side generally covered greater distances across all levels of play," it would be good to compare the numerical data from that study with the present one.
Thank for you comment. We have added ”In this sense Ramón-Llin (2023) compared the distance travelled per point (10.74 m for the player on the right side versus 11.43 m for the player on the left side), while in our study we have compared the total distance per match (3381 m for the player on the right side versus 3476 m for the player on the left side) and the total distance per match (3381 m for the player on the right side versus 3476 m for the player on the left side)”.
- It is suggested to add lower-body eccentric exercises and discuss the application of bilateral and unilateral exercises, considering padel movements: "Conditioning programs for left-side players could focus on agility, explosiveness, and anaerobic endurance to sustain the intense, repetitive bursts of movement required during matches."
We have added this appreciation in the manuscript. “As well as adding lower-body eccentric, bilateral or unilateral exercises, considering the different movements on the track”
Conclusions:
- Again, in the conclusions, it is necessary to mention that the study focuses on external load.
- The conclusions are too lengthy, and some parts belong to the discussion, such as the information starting with "These results align with the understanding." My proposed conclusion to give the authors an idea is as follows:
- "The findings of this study indicate that left-side professional padel players exhibit higher values across most external load variables compared to right-side players. Notably, they showed significantly higher relative acceleration rates per hour, reflecting a greater physical load. Additionally, a trend toward higher relative deceleration values was observed for left-side players.
- These results underscore the importance of position-specific conditioning in professional padel. Coaches and trainers should consider the distinct physical demands of left-side players when designing targeted training programs, ensuring their preparation is tailored to the more active and physically demanding role they assume during matches."
Thanks for your effort helping us. The conclusion are “The study was one of the first to compare novel external loading variables such as player load and acceleration profiles considering position. The findings of this study in-dicate that left-side professional padel players exhibit higher values across most external load variables compared to right-side players. Notably, they showed close to be significantly higher relative acceleration rates per hour, reflecting a greater physical load.
These results underscore the importance of position-specific conditioning in profes-sional padel. Coaches and trainers should consider the distinct physical demands of left-side players when designing targeted training programs, ensuring their preparation is tailored to the more active and physically demanding role they assume during matches.
Comments on the Quality of English Language
The English language in the manuscript is generally clear but could benefit from minor revisions to improve readability and precision.
Thank for you comment. We have made some revisions to improve readability:
Instead of “Inertial measurement devices enable detailed analysis of competition load in paddle tennis by providing kinematic variables that enhance athlete performance across various sports.” we have rewritten to “Inertial measurement devices enable a comprehensive analysis of competitive load in padel by providing kinematic variables that enhance players' performance in this discipline.”
Instead of “Comparative analysis indicated that players positioned on the left side exhibited a higher frequency of accelerations and decelerations per hour than those on the right.” we have rewritten to “Left-side players showed more frequent accelerations and decelerations per hour compared to right-side players.”
Instead of “In recent years, padel has rapidly spread worldwide as a popular new sport, with the professional circuit also expanding internationally. Currently, Premier Padel is the primary official professional padel circuit, featuring 25 tournaments across 18 countries on five continents.” we have rewritten to “Padel has grown rapidly as a global sport, with the Premier Padel circuit hosting 25 tournaments across 18 countries on five continents.”
Instead of “Research on padel has predominantly centred on identifying performance indicators and analysing competition load.” we have rewritten to “Research on padel mainly focuses on performance indicators and competition load.”
Instead of “Prior to the general warm-up, players are fitted with a tracking vest without the Wimu device. Following the completion of specific warm-up exercises, the WimuPro™ device is inserted into the vest, activated, and data recording commences.” we have rewritten to “Before the general warm-up, players were fitted with a tracking vest without the Wimu device. After completing specific warm-up exercises, the WimuPro™ device was inserted into the vest, activated, and data recording began.”
Instead of “Players positioned on the left side demonstrated higher median values than those on the right side across most variables. However, these differences were not statistically significant for most comparisons.” we have rewritten to “Left-side players showed higher median values across most variables, close to significant differences in relative accelerations (Mdnright = 377.54 vs Mdnleft = 413.54; p = 0.05; r = 0.22). Other differences, such as relative decelerations (Mdnright = 333.59 vs Mdnleft = 373.96; p = 0.06; r = 0.20), approached significance”
Instead of “These findings suggest that left-side players may experience greater physical demands due to technical-tactical actions that require increased mobility and broader court coverage. Left-side players are typically more involved in gameplay, executing a greater number of shots such as volleys, smashes, and shots against the wall, in comparison to right-side players.” we have rewritten to “Left-side players face greater physical demands due to technical-tactical actions requiring more mobility and court coverage. They engage more frequently in play, executing more volleys, smashes, and wall rebounds compared to right-side players.”
Instead of “The findings indicate that players positioned on the left side generally displayed higher values across most load-related variables.” we have rewritten to “The findings of this study indicate that left-side professional padel players exhibit higher values across most external load variables compared to right-side players. Notably, they showed close to be significantly higher relative acceleration rates per hour, reflecting a greater physical load”

Reviewer 2 Report
Comments and Suggestions for Authors
Thank you for your submission. After carefully reviewing your manuscript, I find that the paper presents valuable findings, but there are several areas that require substantial revision before it can be considered for publication. I have outlined my specific comments below:
- Structured Abstract:
The abstract needs to be revised to follow a structured format, as outlined in the journal’s submission guidelines. Please restructure the abstract accordingly.
- Reference Formatting:
The reference formatting throughout the manuscript appears to be inconsistent with the journal’s required style. I strongly recommend reviewing the reference list and adjusting it to align with the formatting guidelines specified by the journal. This will ensure that the manuscript meets the submission requirements.
- Ethical Approval Information:
The manuscript lacks a section on ethical approval. Please ensure that you provide the necessary information regarding ethical approval for the study, including the name of the approving ethical review board and the approval number.
- Normality Test Results:
A normality test is mentioned in the statistical methods, but the corresponding results are not provided in the Results section. Please include the results of the normality test (such as p-values or test statistics) to support your statistical analysis.
- Tables and Figures:
Table 1 and Figure 1: Both Table 1 and Figure 1 need to include detailed captions. For example, in Figure 1, please explain the significance of the inner and outer bounding boxes and clarify the meaning of the red and blue dots, indicating that they represent individual sample data.
Table 1: The format of the fifth-to-last row appears inconsistent with the rest of the table. Please adjust the formatting for uniformity.
- Discussion – Limitations:
In the Discussion section, it is important to address the limitations of your study. For example, although the study focuses on external load in athletes, it does not investigate internal load, such as heart rate, which could provide valuable insights. I suggest adding a discussion on these limitations and proposing future research directions, such as investigating kinematic and dynamic parameters in movement techniques, and incorporating internal load measures. Some recently studies shall be added in the discussion, such as: ‘Wearable Movement Data as a Potential Digital Biomarker for Chronic Pain: An Investigation Using Deep Learning’, Physical Activity and Health, 8(1), p. 83–92.
- Title Revision:
The manuscript primarily focuses on external load, and there is no discussion of internal load (e.g., heart rate). To better reflect the content of the paper, I recommend revising the title to: Impact of Playing Position on Competition External Load in Professional Padel Players Using Inertial Devices.
Author Response
REVIEWER 2
We appreciate your comments and improvements to the manuscript in advance. We have made all the pertinent corrections in it, you can see the changes made in the manuscript in red. We have also noted the changes below the comment in this document.
- Structured Abstract:
The abstract needs to be revised to follow a structured format, as outlined in the journal’s submission guidelines. Please restructure the abstract accordingly.
The authors thank you for your comment, we have made the relevant changes to the manuscript.
“Padel is a racket sport that has grown internationally, both in the number of players and in the number of competitions. Inertial measurement devices enable a comprehensive analysis of competitive load in padel by providing kinematic variables that enhance players' performance in this discipline. This study aimed to analyse the external load variables recorded with an inertial device in elite padel players, comparing metrics based on the players’ positions (left and right sides of the court). A total of 83 players were monitored during 23 matches of the professional circuit. The results revealed specific load metrics, including distance covered, frequency of accelerations and decelerations per hour, maximum speeds reached, and acceleration profiles relative to distance covered, all measured using the Wimu Pro™ device. Left-side players showed more frequent accelerations and decelerations per hour compared to right-side players. The results of this study will, on the one hand, enable the adjustment of new specific parameters for professional padel training, such as acceleration and deceleration profiles, player load, and distances covered at explosive speeds. On the other hand, the results will provide a more objective evaluation of padel players' performance based on their positions.”.
- Reference Formatting:
The reference formatting throughout the manuscript appears to be inconsistent with the journal’s required style. I strongly recommend reviewing the reference list and adjusting it to align with the formatting guidelines specified by the journal. This will ensure that the manuscript meets the submission requirements.
We have modified the reference formatting.
- Ethical Approval Information:
The manuscript lacks a section on ethical approval. Please ensure that you provide the necessary information regarding ethical approval for the study, including the name of the approving ethical review board and the approval number.
As indicated in the manuscript and also indicated to the editors of the journal, the data collection process was carried out by the Paddle Federation. Therefore, the University’s ethics committee is not required to approve this research. We have added information “All players participated voluntarily, providing informed consent before the commence-ment of the study. A Regional Padel Federation was responsible for obtaining the neces-sary permits for recording both the facilities and the players. This study adhered to the ethical principles outlined in the Declaration of Helsinki.”
- Normality Test Results:
A normality test is mentioned in the statistical methods, but the corresponding results are not provided in the Results section. Please include the results of the normality test (such as p-values or test statistics) to support your statistical analysis.
Thank for your comment.In doctoral dissertation is common to present Normality test results but not in these kind of papers. We show the Normality test and we have attached the data base in the files
Pruebas de normalidad |
|||||||
|
Posición |
Kolmogorov-Smirnova |
Shapiro-Wilk |
||||
|
Estadístico |
gl |
Sig. |
Estadístico |
gl |
Sig. |
|
Accelerations (count) |
Derecha |
,168 |
41 |
,005 |
,913 |
41 |
,004 |
Revés |
,169 |
42 |
,004 |
,923 |
42 |
,008 |
|
Decelerations (count) |
Derecha |
,168 |
41 |
,005 |
,915 |
41 |
,005 |
Revés |
,170 |
42 |
,004 |
,925 |
42 |
,009 |
|
Distance (m) |
Derecha |
,139 |
41 |
,045 |
,948 |
41 |
,059 |
Revés |
,109 |
42 |
,200* |
,956 |
42 |
,107 |
|
Max Speed (km/h) |
Derecha |
,092 |
41 |
,200* |
,969 |
41 |
,327 |
Revés |
,080 |
42 |
,200* |
,953 |
42 |
,082 |
|
Player Load (a.u.) |
Derecha |
,202 |
41 |
<,001 |
,897 |
41 |
,001 |
Revés |
,216 |
42 |
<,001 |
,910 |
42 |
,003 |
|
Explosive Distance (m) |
Derecha |
,107 |
41 |
,200* |
,965 |
41 |
,237 |
Revés |
,098 |
42 |
,200* |
,952 |
42 |
,076 |
|
HSR Rel (m) |
Derecha |
,290 |
41 |
<,001 |
,532 |
41 |
<,001 |
Revés |
,282 |
42 |
<,001 |
,493 |
42 |
<,001 |
|
[0, 1] |
Derecha |
,189 |
41 |
<,001 |
,910 |
41 |
,003 |
Revés |
,187 |
42 |
<,001 |
,928 |
42 |
,011 |
|
[1, 2] |
Derecha |
,283 |
41 |
<,001 |
,456 |
41 |
<,001 |
Revés |
,422 |
42 |
<,001 |
,247 |
42 |
<,001 |
|
[2, 3] |
Derecha |
,154 |
41 |
,016 |
,947 |
41 |
,054 |
Revés |
,113 |
42 |
,200* |
,937 |
42 |
,023 |
|
[3, 4] |
Derecha |
,126 |
41 |
,103 |
,959 |
41 |
,146 |
Revés |
,135 |
42 |
,054 |
,943 |
42 |
,037 |
|
[4, 5] |
Derecha |
,204 |
41 |
<,001 |
,884 |
41 |
<,001 |
Revés |
,137 |
42 |
,046 |
,929 |
42 |
,012 |
|
[5, 6] |
Derecha |
,403 |
41 |
<,001 |
,661 |
41 |
<,001 |
Revés |
,474 |
42 |
<,001 |
,533 |
42 |
<,001 |
|
[6, 10] |
Derecha |
,540 |
41 |
<,001 |
,226 |
41 |
<,001 |
Revés |
. |
42 |
. |
. |
42 |
. |
|
[-1, 0] |
Derecha |
,185 |
41 |
,001 |
,920 |
41 |
,007 |
Revés |
,186 |
42 |
<,001 |
,925 |
42 |
,009 |
|
[-2, -1] |
Derecha |
,119 |
41 |
,151 |
,926 |
41 |
,010 |
Revés |
,121 |
42 |
,127 |
,937 |
42 |
,022 |
|
[-3, -2] |
Derecha |
,128 |
41 |
,088 |
,941 |
41 |
,033 |
Revés |
,113 |
42 |
,200* |
,966 |
42 |
,237 |
|
[-4, -3] |
Derecha |
,205 |
41 |
<,001 |
,887 |
41 |
<,001 |
Revés |
,132 |
42 |
,063 |
,930 |
42 |
,013 |
|
[-5, -4] |
Derecha |
,186 |
41 |
,001 |
,903 |
41 |
,002 |
Revés |
,214 |
42 |
<,001 |
,860 |
42 |
<,001 |
|
[-6, -5] |
Derecha |
,460 |
41 |
<,001 |
,566 |
41 |
<,001 |
Revés |
,410 |
42 |
<,001 |
,634 |
42 |
<,001 |
|
[-10, -6] |
Derecha |
,540 |
41 |
<,001 |
,226 |
41 |
<,001 |
Revés |
,537 |
42 |
<,001 |
,284 |
42 |
<,001 |
|
Max Acceleration |
Derecha |
,123 |
41 |
,118 |
,957 |
41 |
,120 |
Revés |
,101 |
42 |
,200* |
,969 |
42 |
,296 |
|
Max Deceleration |
Derecha |
,292 |
41 |
<,001 |
,540 |
41 |
<,001 |
Revés |
,135 |
42 |
,054 |
,966 |
42 |
,233 |
|
*. Esto es un límite inferior de la significación verdadera. |
|||||||
a. Corrección de significación de Lilliefors |
- Tables and Figures:
Table 1 and Figure 1: Both Table 1 and Figure 1 need to include detailed captions. For example, in Figure 1, please explain the significance of the inner and outer bounding boxes and clarify the meaning of the red and blue dots, indicating that they represent individual sample data.
Thank you very much for your comment. In the figure below the blue dot plots ‘right’ is indicated to refer to the data of the players on the right side and below the red dot plots ‘left’ is indicated to refer to the data of the players on the left side. In this type of scatterplot, the red dot represents the geometric centre of the point cloud.
Table 1: The format of the fifth-to-last row appears inconsistent with the rest of the table. Please adjust the formatting for uniformity.
Thanks for your comment. We have made the changes in table 1
- Discussion – Limitations:
In the Discussion section, it is important to address the limitations of your study. For example, although the study focuses on external load in athletes, it does not investigate internal load, such as heart rate, which could provide valuable insights. I suggest adding a discussion on these limitations and proposing future research directions, such as investigating kinematic and dynamic parameters in movement techniques, and incorporating internal load measures. The authors agree with your comments regarding the discussion section, with this we have made the changes that you indicate in the manuscript. We have added “It would be very interesting to study how external load evolves when comparing the first and second halves of the match. In our study, we did not perform measurements on professional female padel players, as this will be the subject of future studies, as well as performing internal load measurements, a clear example could be HR. It will also be the purpose of future research to analyse the kinematic and dynamic parameters in movement techniques”.
Some recently studies shall be added in the discussion, such as: ‘Wearable Movement Data as a Potential Digital Biomarker for Chronic Pain: An Investigation Using Deep Learning’, Physical Activity and Health, 8(1), p. 83–92.
We have added that reference in “As well as adding lower-body eccentric, bilateral or unilateral exercises, considering the different movements on the track. Additionally, coaches could use these insights to develop tactical strategies that play to each player’s strengths. Therefore, players on the left side should be managed differently in terms of recovery and rest variables to prevent injuries [47]. These factors will be crucial for coaches to address, tailoring them according to each player's specific load.”
- Title Revision:
The manuscript primarily focuses on external load, and there is no discussion of internal load (e.g., heart rate). To better reflect the content of the paper, I recommend revising the title to: Impact of Playing Position on Competition External Load in Professional Padel Players Using Inertial Devices.
The authors agree with your comment on the title, we have included the changes in the manuscript. Thus title is “Impact of Playing Position on Competition External Load in Professional Padel Players Using Inertial Devices”

Round 2
Reviewer 2 Report
Comments and Suggestions for Authors
All my questions have been well addressed, now I recommend to accpet this paper.
Author Response
Thank you for your comments.